# Interventions to Improve Treatment Outcomes among Adolescents on Antiretroviral Therapy with Unsuppressed Viral Loads: A Systematic Review

**DOI:** 10.3390/ijerph19073940

**Published:** 2022-03-25

**Authors:** Farai Kevin Munyayi, Brian van Wyk, Yolanda Mayman

**Affiliations:** 1School of Public Health, University of the Western Cape, Cape Town 7535, South Africa; bvanwyk@uwc.ac.za; 2Department of Psychology, University of the Western Cape, Cape Town 7535, South Africa; 3570478@myuwc.ac.za

**Keywords:** adolescents, HIV, antiretroviral, retention, viral suppression

## Abstract

Adolescents living with HIV (ALHIV) face unique developmental challenges that increase the risk of unsuppressed viral loads. Current reviews present a need for proven interventions to improve viral suppression among ALHIV on ART, who have a history of unsuppressed viral loads. This systematic review aims to synthesize and appraise evidence of the effectiveness of interventions to improve treatment outcomes among ALHIV with unsuppressed viral loads. Six bibliographic databases were searched for published studies and gray literature from 2010 to 2021. The risk of bias and certainty of evidence was assessed using the ROBINS-I tool, CASP checklists and GRADE. A total of 28 studies were eligible for full-text screening; and only three were included in the qualitative synthesis. In addition, two studies were included from website searches. Four types of interventions to improve viral suppression were identified, namely: intensive adherence counselling; community- and facility-based peer-led differentiated service delivery (DSD); family based economic empowerment; and conditional economic incentives and motivational interviewing. We strongly recommend peer-led community-based DSD interventions, intensive adherence counselling, and family-based economic empowerment as potential interventions to improve viral suppression among ALHIV.

## 1. Introduction

Globally, 1.6 million adolescents who are aged 10–19 years were estimated to be living with HIV in 2018: and approximately 85% of them were residing in sub-Saharan Africa [1,2]. Adolescents are defined by the World Health Organization (WHO) as individuals who are aged between 10–19 years [3]. Initiatives such as the elimination of mother-to-child transmission (eMTCT) program, along with the developments in HIV treatment options for pediatrics, have seen some remarkable successes and more children living with HIV are surviving into adolescence [4]. Newer innovations such as Point-of-Care diagnostics have improved Early Infant Diagnosis, and enhanced prompt initiation of life-saving HIV treatment, and have indeed contributed to the large proportions of perinatally infected children who are growing into adolescence [5]. Furthermore, behaviorally HIV-infected adolescents living with HIV (ALHIV), also contribute to the increases in incidence of HIV, and these may respond differently to interventions that may work for perinatally infected adolescents [6].

Routine data reports and studies have shown that uptake and access of comprehensive HIV care and treatment packages amongst this population group are much lower than in adults [7]. During 2019, it was estimated that 53% (36–64%) of the 0–14 years old children had access to treatment, compared to 68% (54–80%) reach for the ≥15 years individuals who were living with HIV [8]. Generally, public sector health settings in several sub-Saharan Africa countries are inadequately equipped to provide guidance and sufficient support to ALHIV for them to stay engaged in HIV care services and to adhere to prescribed treatment regimens [9]. The WHO defined characteristics of adolescent-friendly health services include that those services should be accessible, appropriate, equitable, acceptable, and effective [10]. Barriers that can be addressed by adolescent-friendly HIV services include having to attend the clinic during school hours, social isolation, fear of disclosure, and conflict with clinic staff, whilst facilitators may include peer support, after school clinic hours, and connection with clinic staff [11].

Continuous engagement in HIV care and good levels of ART adherence are key ingredients to achieving and sustaining viral load suppression, which is in turn essential for ensuring the overall health of the ALHIV [12]. Conversely, inconsistency or interruptions in ART treatment increases the risk of developing drug resistant mutations, which leads to progression to advanced HIV disease. Another consequence of interrupting treatment is diminishing future treatment options and increased risk of unsuppressed viral loads which increase further transmission rates [13]. Despite the reported high levels of ART adherence globally (>95%), the consistently diminishing levels of adherence with time remains a huge concern. Additionally, as HIV programs scale up, rates of loss to follow up have been a concerning phenomenon [14]. Currently, both community and facility-based strategies provide evidence-based interventions to alleviate adherence challenges experienced by individuals taking HIV treatment. The interventions may include individual counseling, motivational interviewing, group education and adherence counselling, pharmacist counselling, mHealth strategies, fast-tracking medication pick-ups, home-based or community differentiated service delivery (DSD) approaches (which includes counselling and treatment support), and financial incentives, including the awarding of disability grants and provision of food [15]. Peer-led support groups, and individual and group adherence counselling interventions, are among the most common interventions for ALHIV.

To date several systematic reviews have reported on the effectiveness of interventions to improve treatment outcomes for ALHIV. The systematic reviews available in the literature identified several promising interventions for improving adherence to antiretroviral treatment and retention in care. However, evidently, there are limited quantities and limitations in quality of the studies with a particular focus on the 10–19 years age group published up until 2015 [16]. MacPherson and colleagues performed a review of ART adherence and retention in care interventions for ALHIV published from 2001 to 2014. The review reported individual and group counseling, financial incentives, youth-friendly services, and increased accessibility to clinics, as some of the effective interventions [17]. An update provided by Casale and colleagues, which reviewed available studies between 2016 and June 2018, found ten relevant studies, mostly from the sub-Saharan Africa region. The review classified the interventions as clinic level, and community or household level interventions, and identified one mHealth intervention. The authors concluded that more multifaceted interventions for ALHIV were needed to address the treatment gaps and socioeconomic barriers for this vulnerable population [16]. For the most part, the interventions have shown mixed results, with the most recent review of adherence interventions for adolescents and youth by Lindsey and colleagues, focusing on low- and middle-income countries, reporting that none of the interventions improved ART adherence or viral load suppression [18].

However, despite some interventions showing promise in improving treatment outcomes for ALHIV, the abovementioned systematic reviews highlight the limited quantity, and low-to-moderate strength of evidence and quality of available studies for ALHIV. These reviews commonly include interventions for all adolescents and youth, despite their viral load suppression levels. Considering the treatment gaps among adolescents, resulting in viral non-suppression, the need to review available evidence on interventions focusing on unsuppressed adolescents becomes urgent. It is crucial to devise, apply, and evaluate interventions focused on attainment and maintaining of viral load suppression among ALHIV. Hence, we conducted a review of studies that strictly evaluated interventions focusing on adolescents who had challenges attaining and sustaining viral suppression. This paper reports on a systematic review of interventions to improve viral suppression among ALHIV with unsuppressed viral loads and provides an assessment of the strength of evidence of effectiveness and makes recommendations for policy and practice.

## 2. Materials and Methods

A protocol has been published which describes the study design and the methods which were used for this systematic review in detail [19]. Additionally, the protocol for this review is registered with the International Prospective Register of Systematic Reviews (PROSPERO) (CRD42021232440). This systematic review was conducted between January to July 2021. The full search strategy is provided in Appendix A.

### 2.1. Study Selection and Data Extraction

The selection of studies for inclusion and multistep screening for eligibility processes were conducted in accordance with the reporting guidance provided in the Preferred Reporting Items for Systematic Reviews and Meta-Analyses Protocol (PRISMA-P) 2015 guidelines [20]. The Mendeley Reference Management Software was used to manage the citations of all the selected studies for screening, and duplicate records were manually removed in Mendeley [21]. The articles’ screening process was conducted by two reviewers (FKM and YM) who independently reviewed titles and abstracts for eligibility, and further independently reviewed the full texts of the selected articles for final inclusion [22]. Disagreements on studies to be included or excluded in the final assessment and analysis were referred for arbitration to a third reviewer (BvW) authorized to resolve any disagreements. A data extraction matrix (see Appendix A) was developed in Microsoft Excel to capture characteristics and key elements of each study that was selected for inclusion in the final review. Only the pre-defined data reported in the articles was extracted although additional information was sought from authors through email for clarification or where such needed data were not available in the report.

### 2.2. Quality Assessment and Risk of Bias

The quality of evidence and risk of bias for the non-randomized control trials and the randomized control trials were assessed using the Risk of Bias in Non-randomized Studies of Interventions (ROBINS-I) Assessment Tool and the Critical Appraisal Skills Programme (CASP) Checklist, respectively [23]. The GRADE considerations were used to assess the certainty of evidence on the interventions and outcomes reported by each included study, considering the risk of bias, consistency in effects, imprecision, indirectness, and publication bias [24,25].

### 2.3. Data Synthesis and Analysis

Quantitative data were presented on the Microsoft Excel (Microsoft Corporation, Washington, DC, USA) data extraction sheet and analyzed descriptively without pooling the measures of effect from each study due to the statistical heterogeneity. The included study interventions and population age ranges were also insufficiently homogeneous to reasonably pool the studies for a meta-analysis. Therefore, qualitative data were categorized as relevant to the research questions and narratively analyzed and reported.

## 3. Results

### 3.1. Identification of Relevant Studies

Our search strategy identified 1664 studies, 894 of which were duplicates and therefore were removed. The screening of titles and abstracts which did not meet the inclusion criteria or clearly irrelevant studies resulted in 28 eligible studies for full text review. Only three studies met the inclusion criteria and therefore qualified for the final review. However, two additional studies, identified through website searches through Google Scholar, also met the inclusion criteria and were added to the final review. The PRISMA flow diagram in Figure 1 describes the results of all searches conducted, the study selection decisions made, and the final studies reported in this review. Twenty-five full text reports that were excluded from the final review comprised of four studies that did not evaluate any intervention, three studies that did not include adolescents aged 10–19 years, eight studies that did not report on viral suppression, six studies that had no disaggregated data for adolescents aged 10–19 years, and four reports that were study protocols.

### 3.2. Study Characteristics

The characteristics of the included studies are summarized in Table 1. All included studies were published between 2018 and 2021; with one paper still in pre-print form [26]. All the included studies were conducted in sub-Saharan Africa (SSA): two in Zimbabwe [27,28] and in Uganda [29,30], and one in Nigeria [26]. Four studies were parallel, or cluster randomized controlled trials [26,27,28,30], and one was a retrospective cohort study [29]. The sample sizes ranged from 134 to 479 participants. Two studies included participants who were younger or older than the 10–19 years age range [27,29]. One study in Uganda included children less than 10 years old whilst another study in Zimbabwe also included young adults aged up to 24 years [27,29].

### 3.3. Description of Interventions

Four types of interventions to improve viral suppression among ALHIV challenged with achieving viral suppression were identified: intensified adherence counselling (IAC) [29], differentiated service delivery (DSD) [27,28], family based economic empowerment [30], and conditional economic incentives with motivational interviewing [26].

The IAC intervention was implemented in Uganda to children and adolescents who had unsuppressed viral loads (≥1000 copies/mL) [29]. It consisted of three sessions of intensive adherence counselling, over three months, by healthcare workers (adherence counsellors or nurses) or trained ‘expert’ patients under the supervision of healthcare workers. ‘Expert’ patients were volunteers who are on ART and had suppressed viral loads. The content of adherence counselling covered the “5 As”—Assess; Advise; Assist; Agree; Arrange—to gain insight on the child/adolescent’s medication administration, adherence barriers, support at home, and the opportunities to enhance adherence to treatment. Older adolescents received counselling on their own; while children/adolescents between 6 and 15 years were accompanied by their caregivers. For children under six years the sessions were conducted with their caregivers only.

The DSD intervention was evaluated in two studies in Zimbabwe [27,28]. The intervention involved assigning a Community Adolescent Treatment Supporter (CATS) to adolescents on ART. The CATS was a trained peer counsellor—an adolescent or a young adult aged 18–24 years, who was living with HIV—who received oversight (supervision) from a program mentor through weekly supervision meetings and peer-to-peer support via Skype or WhatsApp group. The CATS also provided continuous individual level HIV care support for adolescents within the community settings. Each CATS provided support to a maximum of 10 adolescents on ART within their catchment area, through text reminders/mobile health, phone calls, home visits, and adherence counselling at the health facility. The target groups for intervention differed by age (13–19 vs. 10–24 years) and cut off for viral suppression (1000 vs. 400 copies/mL). In addition, all study participants received standard care for ART which comprised of routine adherence support by nurses and adult counsellors (individual adherence counselling and group sessions), three monthly clinic visits, pill counts, adherence consultations, prescription refills, and six monthly CD4 count monitoring.

The family-based economic empowerment intervention and the conditional economic incentives and motivational intervention provided financial incentives to promote behavior that would lead to viral suppression in adolescents who had unsuppressed viral loads. The family-based economic empowerment intervention was implemented in Uganda, targeting adolescents aged 10–16 years [30]. The intervention consisted of the provision of incentivized savings’ accounts for adolescents, for medical- and education-related expenses and microenterprise promotion. The microenterprise workshops consisted of four one-hour group training sessions on microenterprise development and financial management for adolescents and their caregivers, and 12 group sessions on business development, goal setting, and avoiding risk.

The conditional economic incentives and motivational interviewing intervention was implemented in Nigeria to adolescents aged 10–19 years old who had unsuppressed viral loads [26]. Participants received US$ 5.6 if they achieved a viral load of less than 20 copies/mL for the first three months; then received US $2.8 if they maintained viral suppression for the next three months, and then again for the next six months. They further received an additional US $5.6 if their viral load remained suppressed for the next 12 months. In addition, attendance of motivational interviewing (MI) sessions at each clinic visit were mandatory to receive the cash incentive. The MI sessions were conducted by adherence counsellors trained in MI techniques, which included risk assessments, risk reduction counselling, and inspiring behavior change [31].

### 3.4. Primary Outcomes of Interventions

Viral Load Suppression

All the five included studies (Table 1) provided data regarding viral suppression levels among adolescents who had challenges achieving and maintaining viral suppression. The IAC intervention in Uganda showed greater proportions of viral suppression among the younger adolescents aged 10–14 years after receiving IAC (25.6% unsuppressed vs. 41.3% suppressed), whereas the suppression proportions among the older adolescents after receiving IAC were at 16.1% unsuppressed vs. 12.7% suppressed [29]. The overall viral suppression levels for all adolescents aged 10–19 years after receiving IAC sessions was low, at 29%.

The two RCTs conducted in Zimbabwe both focused on the peer-led intervention known as the Zvandiri intervention, although the implementation models were different. The peer-led multicomponent DSD intervention enrolled participants who were not virally suppressed at 1000 copies/mL at 16 rural primary healthcare clinics [28]. The study reported a protective effect of the intervention on viral suppression (risk ratio = 1.17 (95% CI 1.04–1.32) compared to standard care.

The other study implemented a community-based peer support intervention at a referral hospital for adolescents who were not virally suppressed at 400 copies/mL [27]. The study reported no difference in the probability of having undetectable viral load between the community-based peer support intervention and standard of care (odds ratio = 1.14 (95% CI 0.82–1.59).

The family-based economic empowerment intervention in Uganda reported higher incidence of undetectable viral loads compared to standard care (incidence rate ratio = 1.47 (95% CI 1.06–2.04) [30].

The two-year study in Nigeria on adolescents with a detectable viral load at >20 copies/mL, reported higher rates or prevalence of suppression levels among adolescents who received conditional economic incentives and motivational interviewing compared to those in standard care. Adolescents who received the intervention had a 10.1% increase in viral suppression whilst there was a 1.6% decrease in the standard of care group over 12 months (a difference of 11.7%, *p* = 0.15) [26].

### 3.5. Secondary Outcomes

Retention in HIV care

The peer-led multicomponent DSD intervention in Zimbabwe showed no difference in retention in HIV care at 96 weeks between the intervention group and those in standard care, with an adjusted prevalence ratio of 0.68 (95% CI 0.23–1.99, *p* = 0.45) [28].

Adherence to ART

Two studies reported adherence to ART. The community-based peer support intervention study reported no difference on self-reported adherence levels at 36 weeks among participants in the intervention arm compared to those in standard care (66.0% vs. 68.9%, *p* = 0.655) [27]. The peer-led multicomponent DSD intervention measured adherence to the scheduled clinic visits and reported no difference between the intervention and standard care groups, with an adjusted prevalence ratio of attendance of <80% of scheduled visits at 0.80 (95% CI 0.32–2.02, *p* = 0.62) [28].

Cost analysis

Two studies [26,28] reported on the respective cost analyses of the interventions. The peer-led multicomponent DSD intervention cost analysis pointed to an increase of three times the cost of standard of care treatment for adolescents receiving HIV treatment [28]. Whilst the annual cost per virally suppressed adolescent is estimated as $450.36 at the standard of care clinics, it was estimated to be $1340.00 for adolescents receiving the intervention. However, the authors argued that the benefits would outweigh these costs, and potential economies of scale could lower the unit costs [28].

The conditional economic incentives and motivational interviewing intervention study reported that the routine care cost was $170.30 per adolescent per annum, whilst the intervention cost was a total of $356.70 [26].

### 3.6. Quality Assessement of Studies

Table 2 shows the risk of bias assessment for the IAC study in Uganda [29]. The overall risk of bias was scored as moderate due to the missing data from the included participants who did not complete the recommended three IAC sessions. Only 77% of the included participants had outcome data after completing the three IAC sessions.

The risk of bias assessments’ results for the four RCTs included in the final analysis are presented in Table 3. Overall, all four studies had a valid study design for an RCT. However, one study in Zimbabwe used a modified intention to treat analysis [28] whilst the other Zimbabwe study amended the eligibility criteria due to slow initial enrolment [27]. The attrition rate in the Uganda RCT necessitated censoring of the participants who were lost to follow-ups and therefore could not have their viral load tests taken [30]. Bias concerns in the Nigerian study included lack of information on whether all recruited participants were accounted for in the final analysis [26].

Given the nature of the interventions implemented in all of the four RCTs, it would not have been possible to blind all the participants, investigators and assessors/analyzers. Another concern was about the different levels of care in the Nigeria study where the adolescents in the intervention group had four viral load tests per year whilst the standard care group had one [26]. There is insufficient detail on the effect of the intervention and the precision of the estimate of the intervention is not clear. Additionally, one of the studies in Zimbabwe was terminated before the planned endpoint due to funding constraints [27].

### 3.7. GRADE Recommendations

The assessments of the certainty of evidence using the GRADE system are presented in Table 4. The results indicate high confidence that the peer-led multicomponent DSD intervention as implemented in Zimbabwe, improves viral suppression among adolescents with unsuppressed viral loads, and the true effect lies close to that of the estimate of the effect reported in the study [28]. We are moderately confident that community-based peer support [27] and family-based economic empowerment interventions [30] improve viral suppression in adolescents with unsuppressed viral loads. On the other hand, our results show low level of confidence in the effectiveness of the IAC intervention, and the conditional economic incentives and motivational interviewing intervention on viral suppression in adolescents with unsuppressed viral loads based on the studies conducted in Uganda [29] and Nigeria [26].

## 4. Discussion

### 4.1. Intensified Adherence Counselling

Intensified Adherence Counselling, also known as Enhanced Adherence Counselling (EAC), has been recommended by the WHO for clients with high viral loads based on recommendations from a systematic review that showed 70% re-suppression after receiving adherence interventions [29]. However, due to the limitations of the IAC intervention in Uganda reported in this review, our confidence in the effect reported from the study on improving viral suppression in adolescents is low [29].

There is limited literature about the effectiveness of EAC in routine HIV care among adolescents on treatment with unsuppressed viral loads. Findings of a study in Swaziland state that children and adolescents had higher risk of virologic failure even after receiving EAC and recommended finer defining of elements of optimal EAC support to achieve the desired outcomes [32]. The Uganda study in our review concluded that viral suppression rates were low (23%) among the children and adolescents who were on treatment and having virologic failure. Compliance with the intervention may have contributed to the low efficacy as it was reportedly poor with only 50% of participants completing the IAC sessions within 200 days instead of 90 days [29].

### 4.2. Peer-Led Multicomponent Differentiated Service Delivery

This model provided the strongest evidence of effectiveness of any intervention based on the GRADE system. There is growing enthusiasm in the potential observed in one-on-one peer-driven support for adolescents receiving ART, on treatment outcomes, particularly retention in HIV care and viral suppression [33]. Adolescence is a critical developmental stage where positive peer influence and support has shown to be an essential component [34]. Results show that the community-based peer support intervention only provided moderate certainty of evidence, with the intervention showing no effect in improving viral suppression among unsuppressed adolescents. This finding supports the argument for tailor-made interventions for adolescents at individual levels, which can be enhanced with group-based components of a comprehensive support package [35]. Although peer support successes are based on shared or similar experiences that facilitate knowledge sharing for practical, social, and emotional support, taking responsibility for assisting others in a similar situation may increase the vulnerability of the peer supporters [33]. Recent evidence shows that peer support programs are being implemented widely but are not adequately described and effectiveness is unknown. Therefore, implementation science research on peer support is needed to evaluate the applicability and effectiveness of DSD models among adolescents [36]. Despite the evidence of expansion of peer support and patient-centered DSD models, a situational analysis of DSD models for adolescents and young people living with HIV in South Africa reported low uptake by, and accessibility to, adolescents, which may indicate that the existing DSD models may not be tailored to the preferences of the adolescents [37].

Consequently, community-based peer support interventions for children, young people and ALHIV have not provided consistent evidence of effectiveness, fueling the debate around applicability of DSD models for adolescents [33]. Barriers in the adolescent HIV care continuum are multifactorial, thus DSD models consisting of multicomponent packages may have the greatest potential to address the multiple and unique needs of adolescents [38]. However, some key considerations must be made in tailoring services to different populations. For example, there are debates around the risks and benefits of enrolling adolescents in DSD models providing multi-month dispensing (MMD) of antiretroviral medications, for example, should MMD or 3–6 months visit spacing be offered to stable adolescents only, and whether DSD eligibility criteria should include adolescents above 15 years only [39]. To address some of these barriers, a case has been made for family-based approaches in DSD models, where adolescents are seen with their adult family members for their clinic reviews and ART refills [40].

Notwithstanding the multiple barriers in delivering DSD models to adolescents, the involvement of peers, trained in providing support to adolescents struggling with viral suppression, through the CATS (peer-to-peer support), and use of mobile health meant the peer-led multicomponent DSD intervention had more reach to a younger population [28]. Although the peer-led multicomponent DSD intervention increased the cost of providing HIV treatment to adolescents three-fold, the authors argued that the benefits of better viral suppression outcomes outweigh these costs in the long run, and the program could be delivered in a resource-constrained, real-world setting, and therefore can be scaled up in similar settings [28].

### 4.3. Family-Based Economic Empowerment

Grimsrud and colleagues argued that, although family-centered support within HIV services provision has been promoted in recent years, the potential benefits have not been fully realized [40]. According to the GRADE system, the family-based economic empowerment intervention provided moderate evidence that the intervention improves viral suppression in unsuppressed adolescents. Grounded in asset theory, family economic empowerment interventions foster household financial stability and promote financial literacy and income generating projects, that in turn mitigate against the development of depression, worse education outcomes, and engagement in risky sexual behavior [41]. A family-based intervention for ALHIV aged 10–13 years and their families in South Africa (VUKA) reported improved youth behavior, mental health, communication, stigma, HIV treatment literacy, and adherence to ART among other dimensions [42].

Another study in Uganda reported improved ART adherence associated with family cohesion and social support from caregiver/family [43]. Comparatively, there was no association between adherence and social support from teachers, classmates, or friends, suggesting that promoting family support and strengthening family relations with ALHIV can be essential in addressing ART adherence challenges among ALHIV [43]. The findings from the family-based economic empowerment intervention reinforced the potential role of financial support or economic interventions in improving HIV treatment outcomes. However, further exploration of the mechanisms through which the family-based economic empowerment intervention works is recommended [30].

### 4.4. Conditional Economic Incentives and Motivational Interviewing

Despite growing evidence, very little is known in terms of the utility of financial incentives on improving ART adherence among ALHIV [44]. The conditional economic incentives and motivational interviewing intervention (Incentive Scheme) provided low evidence of effectiveness on viral suppression for unsuppressed adolescents. The authors argued that monetary rewards would likely be attractive for adolescents and struggling young individuals and influence their behaviors, while the lure of the monetary rewards also provides an opportunity to engage with a healthcare provider and increase the chances of achieving better treatment outcomes [26]. An experiment conducted in Cape Town between 2017–2019 identified the incentive amount, incentive format, recipients, delivery mode, and program participants as key components [44]. A combined economic empowerment and peer support intervention in rural Rwanda achieved successes in high pharmacy attendance, attaining target savings, and in viral suppression [45].

A scoping review of psychosocial support interventions to improve adherence and retention in HIV care for young people living with HIV identified four HIV care modalities namely, support groups, family-centered services, treatment supporters, and individual counselling. Motivational interviewing (MI) was an approach that influenced individual positive health behavior change and in improving viral suppression [46]. MI has long been recognized as an effective counselling technique for facilitating behavior change, and in recent times has been integrated into mobile health (mHealth) interventions, beyond the commonly applied in-person models, with promising results [47]. However, the findings from the combined conditional economic incentives and MI (Incentive Scheme) concluded that the intervention was cost-effective and had potential to improve viral suppression and retention in care rates for ALHIV. Although the intervention was highly acceptable among patients, the healthcare workers had concerns about the long-term sustainability of such an intervention [26]. The authors reiterated that governmental and all stakeholders’ involvement and support needs to be guaranteed for the intervention to be implemented in a sustainable manner.

### 4.5. Strengths and Limitations of the Review

To our knowledge, this is the first systematic review to examine the interventions specifically targeting adolescents with unsuppressed viral loads. This study contributes to the body of knowledge around available and evidence-based interventions targeting adolescents with unsuppressed viral loads. The results could inform adolescent-friendly HIV care services and guidelines in both community and healthcare facility settings. The review utilized the updated standardized Preferred Reporting Items for Systematic reviews and Meta-Analyses Protocols (PRISMA) 2020 guidelines. Our search strategy included six major electronic databases with peer-reviewed articles, and we conducted additional website searches for gray literature.

The potential limitations of this study include the restriction to studies published in the English language only, which may have led to omission of potentially relevant studies published in other languages. Another limitation of this study may be related to the timeline of the search strategy (2010–2020), although we conducted an additional internet search up to July 2021.

## 5. Conclusions

This review demonstrated that there is a paucity of interventions addressing treatment outcomes in ALHIV who are virally unsuppressed. However, there is high confidence in the effectiveness of a peer-led multicomponent DSD intervention whilst there is low to moderate quality evidence that intensified adherence counselling, family-based economic empowerment, conditional economic incentives, and motivational interviewing improve viral suppression in ALHIV with unsuppressed viral loads.

## Figures and Tables

**Figure 1 ijerph-19-03940-f001:**
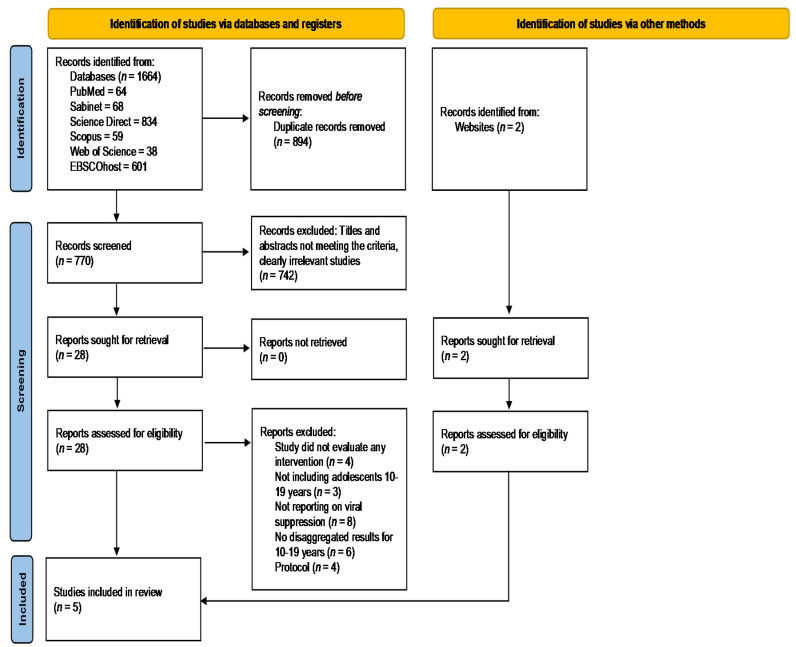
PRISMA 2020 flow diagram.

**Table 1 ijerph-19-03940-t001:** Summary characteristics of included studies of interventions.

First Author, Year	Study Country, Settings	Study Design	Study Population, Sample Size	Intervention	Follow-Up Period	Outcome(s) Measured	Results
Nasuuma, 2018	Uganda, 15 Public health facilities	Retrospective Cohort	9 months–19 years, *N* = 449; *n* = 192 adolescents (10–19 years)	Intensified Adherence Counselling: - children and adolescents with VL ≥ 1000 copies/mL. - 3 sessions; 1 per month - Provided by healthcare workers (counsellors and Nurses) and trained expert patients	19 months	Viral Suppression (<1000 copies/mL) after 3 monthly sessions	Overall viral suppression (10–19 years): 29%
Mavhu, 2020	Zimbabwe, 16 PHC clinics	Cluster RCT	13–19 years, *N* = 496	Peer-led multicomponent DSD intervention (Zvandiri): Enhanced HIV care support for unsuppressed adolescents VL ≥ 1000 copies/mL through: - Community Adolescent Treatment Supporter (CATS), through: Text reminders, calls, home visits and clinic-based counselling, monthly support group, in collaboration with Zvandiri mentors, Nurses and Primary care counsellors - Caregivers were also invited to a 12-session monthly support group Control group: Standard care (adherence counselling for unsuppressed clients).	96 weeks	Viral Suppression (<1000 copies/mL)	Risk Ratio = 1.17 (95% CI 1.04–1.32)
						Retention in Care	Adjusted prevalence ratio of discontinuation of ART for ≥3 months = 0.68 (95% CI 0.23–1.99, *p* = 0.45)
						Adherence	Adjusted prevalence ratio of attendance <80% of scheduled visits = 0.80 (95% CI 0.32–2.02, *p* = 0.62)
Ssewamala, 2020	Uganda, 39 Healthcare clinics, 5 districts	Cluster RCT	10–16 years, *N* = 702; *n* = 288 (adolescents with detectable VL at baseline)	Family-based economic empowerment intervention. Intervention group received: - Incentivized savings accounts for adolescents for medical and education related expenses. - Microenterprise promotion/workshops (four one-hour group sessions on microenterprise development and financial management training for adolescents and their caregivers, and 12 group sessions on business development, goal setting, and avoiding risk. Control group: Standard of care (SOC), medical and psychosocial support (ART and adherence information leaflets, adherence sessions facilitated by lay counsellors and expert clients who are living with HIV)	5 years	Viral Suppression (<40 copies/mL)	Incidence Rate Ratio = 1.468 (95% CI 1.064–2.038, *p* = 0.008).
Ndhlovu, 2021	Zimbabwe, 1 Referral Hospital, Family Care Clinic	RCT	10–24 years, *N* = 212; *n* = 134 (63%) adolescents aged 10–19 years	Community-based peer support intervention (Zvandiri): Enhanced HIV care support for unsuppressed adolescents VL ≥ 400 copies/mL through: - Community Adolescent Treatment Supporter (CATS), through: Text reminders, calls, home visits and clinic-based counselling, monthly support group, in collaboration with Zvandiri mentors, Nurses and Primary care counsellors - Caregivers were also invited to a 12-session monthly support group Control group: Standard of care (adherence counselling for unsuppressed clients).	36 weeks	Viral Suppression (<1000 copies/mL)	Adjusted OR = 1.14 (95% CI 0.82–1.59), *p* = 0.439, at week 36
						Self-reported Adherence (≥95%)	Intervention arm = 66.0% SOC arm = 68.9% *p* = 0.655, at week 36
Ekwunife, (Pre-print)	Nigeria, 12 Hospitals	Cluster RCT	Adolescents (10–19 years), *N* = 246	Conditional Economic Incentives and Motivational Interviewing Intervention group received: - Financial incentives for achieving and maintaining viral suppression <20 copies/mL. - The cash reward was linked to attending motivational interviewing (MI) sessions at each clinic visit. - Monitored by designated nurse - In addition to SOC, MI session at baseline and following ART initiation, by trained counsellor Control group: Standard of care, - monthly or every two months’ scheduled clinic visits, prescription refills - adherence counselling - viral load assessment twice a year, and - annual CD4 counts	2 years	Viral Suppression VL < 20 copies/mL	The difference in viral suppression between intervention and control group = 11.7%

**Table 2 ijerph-19-03940-t002:** Rise of bias assessment (non-randomized studies using ROBINS-1 tool).

ROB Domain	Nasuuma et al., 2018
Bias due to confounding	Low
Bias in selection of participants into the study	Low
Bias in classification of interventions	Low
Bias due to deviations from intended interventions	Moderate *
Bias due to missing data	Moderate *
Bias in measurement of outcomes	Low
Bias in selection of the reported result	Low
Overall Risk of Bias	MODERATE

* Scored as moderate due to only 77% of the included participants having outcome data after completing three Enhanced Adherence Counselling sessions.

**Table 3 ijerph-19-03940-t003:** Risk of Bias Assessment (Randomized trials using CASP tool).

	Mavhu et al., 2020	Ssewamala et al., 2020	Ndhlovu et al., 2021	Ekwunife et al., Pre-Print, 2021
Section A: Is the basic study design valid for an RCT?				
1. Did the study address a clearly focused research question?	√	√	√	√
2. Was the assignment of participants to interventions randomized?	√	√	√	√
3. Were all participants who entered the study accounted for at its conclusion?	√	√	√	Cannot tell
Section B: Was the study methodologically sound?				
4. Were participants, investigators, assessors/analyzers “blinded”?	X	X	X	X
5. Were the study groups similar at the start of the RCT?	√	√	√	√
6. Apart from experimental intervention, did each group receive same level care?	√	√	√	X
Section C: What are the results?				
7. Were the effects of intervention reported comprehensively?	√	√	√	Cannot tell
8. Was the precision of the estimate of the intervention/treatment effect reported?	√	√	√	X
9. Do the benefits of the experimental intervention outweigh the harms/costs?	√	√	√	√
Section D: Will the results help locally?				
10. Can the results be applied to your local population/in your context?	√	√	√	√
11. Would the experimental intervention provide greater value to people in your care?	√	√	√	√

**Table 4 ijerph-19-03940-t004:** GRADE (certainty of evidence).

Domain	Nasuuma et al., 2018	Mavhu et al., 2020	Ssewamala et al., 2020	Ndhlovu et al., 2021	Ekwunife et al., Pre-Print, 2021
Risk of Bias	Moderate	Low	Low	Low	High
Consistency	NA	NA	NA	NA	NA
Directness	High	High	High	High	Low
Imprecision	NA	Low	Moderate	Moderate	NA
Publication Bias	NA	NA	NA	NA	NA
Final Quality of Evidence	LOW ^d^	HIGH	MODERATE ^a^	MODERATE ^e^	LOW ^o^

^a^ Downgraded by 1 due to imprecision, wide 95% Confidence Interval (CI); ^e^ Downgraded by 1 due to incomplete data, study not completed as planned, wide 95% CI which also includes 1; ^d^ Downgraded to low quality of evidence based on the non-randomized study design; ^o^ Downgraded by 2 points due to high Risk of Bias, no CIs reported and missing information.

## Data Availability

Not applicable.

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
