# Peer review of "Interventions to Improve Treatment Outcomes among Adolescents on Antiretroviral Therapy with Unsuppressed Viral Loads: A Systematic Review"

_ijerph, 2022, doi:10.3390/ijerph19073940_

Round 1

Reviewer 1 Report

This study systematically reviewed the interventions to improve treatment outcomes among adolescents on antiretroviral therapy with unsuppressed viral loads. It is an interesting and well-organized study, which contributes to presenting the intervention condition in the relative area with a solid methodological foundation. The conclusions are consistent with the evidence and arguments presented. In my viewpoint, this current manuscript addressed the main research question posed, and is almost ready for publication, with some minor revisions to make. 
First, I suggest the authors better present the introduction part in a logical way, the current presentation is a bit confusing to me why the authors want to target this research problem and how it is innovative. 
Second, please double-check the manuscript for possible typo errors and spelling mistakes.

Reviewer 2 Report

I found this review useful to the readers that focuses on the interventions on HIV adolescents.

The studies are shown clearly adopting tables and the studies are discussed giving a value added approach to this topic. 

I have no concerns on the manuscript. I found it clear, methodologically robust and with signifcant indications for clinical interventions.

Author Response

Thank you so much for reviewing our manuscript and for your valuable feedback on the value added to this important topic. We are glad that you found the paper to be clear, methodologically and clinically sound. 

Reviewer 3 Report

The authors of this article addressed a very important topic on HIV care and viral suppression in adolescents living with HIV by reviewing available data on interventions. Control of HIV viral load is crucial in HIV care, particularly in adolescents, and interventions are lacking.

Overall, the manuscript is well written and provides a good summary of the interventions currently in practice, particularly in Africa where the burden of HIV is greatest.

Minor changes/suggestions

  1. Line 43 to 45, this sentence is not clear. Please simplify and avoid the use of a percentage of percentage for easy understanding.
  2. Line 144 to 147, please state the reasons for excluding 25 studies out of 28 eligible studies.
  3. Table, try to include reasons for exclusion at each step. One would like to know how did you decide to exclude these eligible studies in order to form an opinion on your methodology and generalizability of your findings.
